# Fault rock heterogeneity can produce fault weakness and reduce fault stability

John D. Bedford [1✉], Daniel R. Faulkner [1] & Nadia Lapusta [2,3]

Geological heterogeneity is abundant in crustal fault zones; however, its role in controlling the mechanical behaviour of faults is poorly constrained. Here, we present laboratory friction experiments on laterally heterogeneous faults, with patches of strong, rate-weakening quartz gouge and weak, rate-strengthening clay gouge. The experiments show that the heterogeneity leads to a significant reduction in strength and frictional stability in comparison to compositionally identical faults with homogeneously mixed gouges. We identify a combination of weakening effects, including smearing of the weak clay; differential compaction of the two gouges redistributing normal stress; and shear localization producing stress concentrations in the strong quartz patches. The results demonstrate that geological heterogeneity and its evolution can have pronounced effects on fault strength and stability and, by extension, on the occurrence of slow-slip transients versus earthquake ruptures and the characteristics of the resulting events, and should be further studied in lab experiments and earthquake source modelling.

[1] Rock Deformation Laboratory, Department of Earth, Ocean and Ecological Sciences, University of Liverpool, Liverpool, UK. [2] Department of Mechanical and Civil Engineering, Division of Engineering and Applied Science, California Institute of Technology, Pasadena, CA, USA. [3] Seismological Laboratory, Division of Geological and Planetary Sciences, California Institute of Technology, Pasadena, CA, USA. ✉email: jbedford@liverpool.ac.uk

Many large crustal faults have been shown to be frictionally weak[1–6] when compared to laboratory measurements of quasi-static fault friction. The coefficient of friction $\mu = \tau / \bar{\sigma}_n$, where $\tau$ is the shear stress during slip and $\bar{\sigma}_n$ is the effective normal stress, of most geological materials is typically measured in the laboratory to be between 0.6 and 0.85 at slow slip speeds, independent of rock type[7], with the exception of a few weak minerals, predominantly phyllosilicates[7,8]. Possible explanations for weak faults in nature, where the apparent $\mu$ at which faults operate is often <0.5, include localization of weak minerals along structural foliations[9–13], dynamic weakening during seismic slip[14], and elevated pore fluid pressure interpreted as lower friction coefficients[15,16]. As well as being apparently weak, many crustal faults also exhibit a spectrum of slip behaviour, with earthquake slip and aseismic creep often occurring on the same fault[17,18] and slow slip phenomena being prevalent at all crustal depths[19]. While the apparent weakness of faults and spectrum of slip behaviour can be attributed to the effects of spatially varying and temporally evolving confinement, temperature, and pore fluid pressure, it is clear that heterogeneity in fault zone rocks (Fig. 1a) can also play an important[20,21], if not dominant, role.

Geological investigations have shown that heterogeneity in fault zone rocks occurs over many different scales, from submillimetre-scale structural foliations[9,10], centimetre- to meter-scale blocks within a shear zone mélange[22], hundreds-of-meters scale where lenses of damaged protolith can be entrapped within the core of wide (km-scale) fault zones[23,24] (e.g. Fig. 1a), to tens-of-kilometers scale variations in rock types[10,25]. The role of large-scale fault rock heterogeneity has been highlighted in a number of studies; for example, it has been suggested that heterogeneities such as seamounts can act as earthquake nucleation sites and control the seismogenic behaviour of subduction zone megathrust faults[26,27]. However, the importance of small-scale fault rock heterogeneity in controlling fault slip behaviour, average fault strength, and fault stability is still uncertain.

Here, the effect of fault rock heterogeneity on fault strength and slip behaviour is investigated by a series of laboratory friction experiments on simulated laterally heterogeneous faults. The faults consist of different sized patches of strong, rate-weakening quartz, and weak, rate-strengthening clay fault gouges. Until now, the majority of previous experimental investigations have been performed using mixtures of different fault gouge materials with varying frictional properties, where the materials are homogeneously mixed together[28–31]; intact wafers of natural gouge have also been used[9,10]. In this work, experiments are performed on both homogeneously mixed and spatially heterogeneous gouge layers consisting of quartz, frictionally strong and rate-weakening, and kaolinite clay powder, frictionally weak and rate-strengthening. The fault gouge layers (50 mm long, 20 mm wide, with a thickness of ~1 mm at the onset of shear, after initial pressurization) are sheared in a direct-shear arrangement (Fig. 1b, see also Supplementary Fig. 1) within a triaxial deformation apparatus (see "Methods"). The heterogeneous gouge layers are constructed by placing different sized patches of fine-grained quartz and clay powder (both <5 μm grain size) adjacent to each other in a symmetrical pattern, with a central quartz patch being bound by two clay patches (Fig. 1b). This symmetrical arrangement ensures that no misalignment between the direct-shear forcing blocks would occur as a result of any differential compaction between the different materials; furthermore, the amount of gouge material used (measured by weight prior to the experiment) was calculated so that the thickness of the quartz and clay gouges were the same after initial pressurization and a small amount of shear (Supplementary Fig. 2). The normal stress is applied by the confining pressure ($P_c$) in the triaxial apparatus, held constant at 60 MPa for all tests in this study, and the pore fluid

pressure ($P_f$) within the gouge is servo-controlled at a constant value of 20 MPa, resulting in the effective normal stress $\bar{\sigma}_n = 40$ MPa ($\bar{\sigma}_n = P_c - P_f$). The gouge layers are sheared up to a maximum displacement of 8.5 mm (shear strain ≈ 10, given the final layer thickness of ~0.85 mm). Monitoring the evolution of shear stress while applying velocity steps from 0.3 to 3 μm·s⁻¹ and back allows the experiments to quantify the rate-and-state friction parameters that determine the stability of fault slip[32]. These sliding velocities are sufficiently slow, given the gouge permeability, to ensure that pore pressure transients do not build up within the gouge layer during shearing[33]. The sizes of the strong yet unstable quartz and weak but stable clay patches are varied to investigate the role of different scales of heterogeneity on the magnitude and stability of fault friction.

## Results

**Fault strength evolution.** The experimental results indicate pronounced differences between the behaviour of laterally heterogeneous faults compared to the laterally homogeneous faults with mixed gouge (Fig. 1). All experiments are characterised by an initially rapid increase in shear stress during the loading phase, before the samples clearly yield—i.e., shear inelastically—after ~1 mm of displacement. After that, the friction coefficient $\mu$ of the homogeneously mixed gouge layers remains relatively constant (Fig. 1d), with rate-and-state effects consistent with results from previous experimental studies[29–31]. In contrast, the heterogeneous gouge layers all show ubiquitous weakening (Fig. 1c), with $\mu$ evolving towards the value of the weaker clay phase. To ensure that the observed weakening was not caused by the arrangement of the different gouge patches in the experiments, tests were performed where the symmetry of the heterogeneous layers was reversed (i.e. a central clay patch bound by two quartz patches). These tests also exhibit similar weakening (Supplementary Fig. 3) suggesting that it is the heterogeneity itself, not the arrangement of the different materials, that causes the weakening. Stable sliding is observed for all homogeneously mixed faults and the majority of heterogeneous faults. However, when the quartz patch in the heterogeneous layers comprises ≥80% of the total sliding area, unstable stick-slip sliding emerges, typically triggered by up-steps in the sliding velocity (Fig. 1c).

The observed weakening of the heterogeneous faults is greater than can be explained by the observed smearing of the clay patches. Microstructural analysis of a heterogeneous layer recovered at the end of an experiment (Fig. 2a) shows smearing of clay into localized boundary Y-shears that propagate into the quartz patch. With progressive smearing and localization of the clay phase (Fig. 2b), the strength of the layer overall is expected to decrease as a greater proportion of the slipping surface can be located within the weak clay phase[34]. As the frictional strength of the endmember gouge compositions is known (i.e. 100% quartz and 100% clay in Fig. 1c), the predicted weakening due to smearing can be calculated (Fig. 2c) by assuming that the overall strength is determined by the strength of the two gouges acting in series, based on their relative proportions (the arithmetic mean of $\mu$, based on the proportions of clay and quartz within the layer). The predicted weakening, associated with the relative increase in length of the clay patches, is considerably less than the observed weakening in the experiments (Fig. 2c), suggesting that clay smearing alone is not responsible for the progressive weakening of heterogeneous faults.

An additional cause of the weakening could be differential compaction between the different gouge materials resulting in a redistribution of normal stress (see Supplementary Note 1 for full discussion of this effect). The volumetric strain data from the endmember quartz and clay gouge experiments show that the

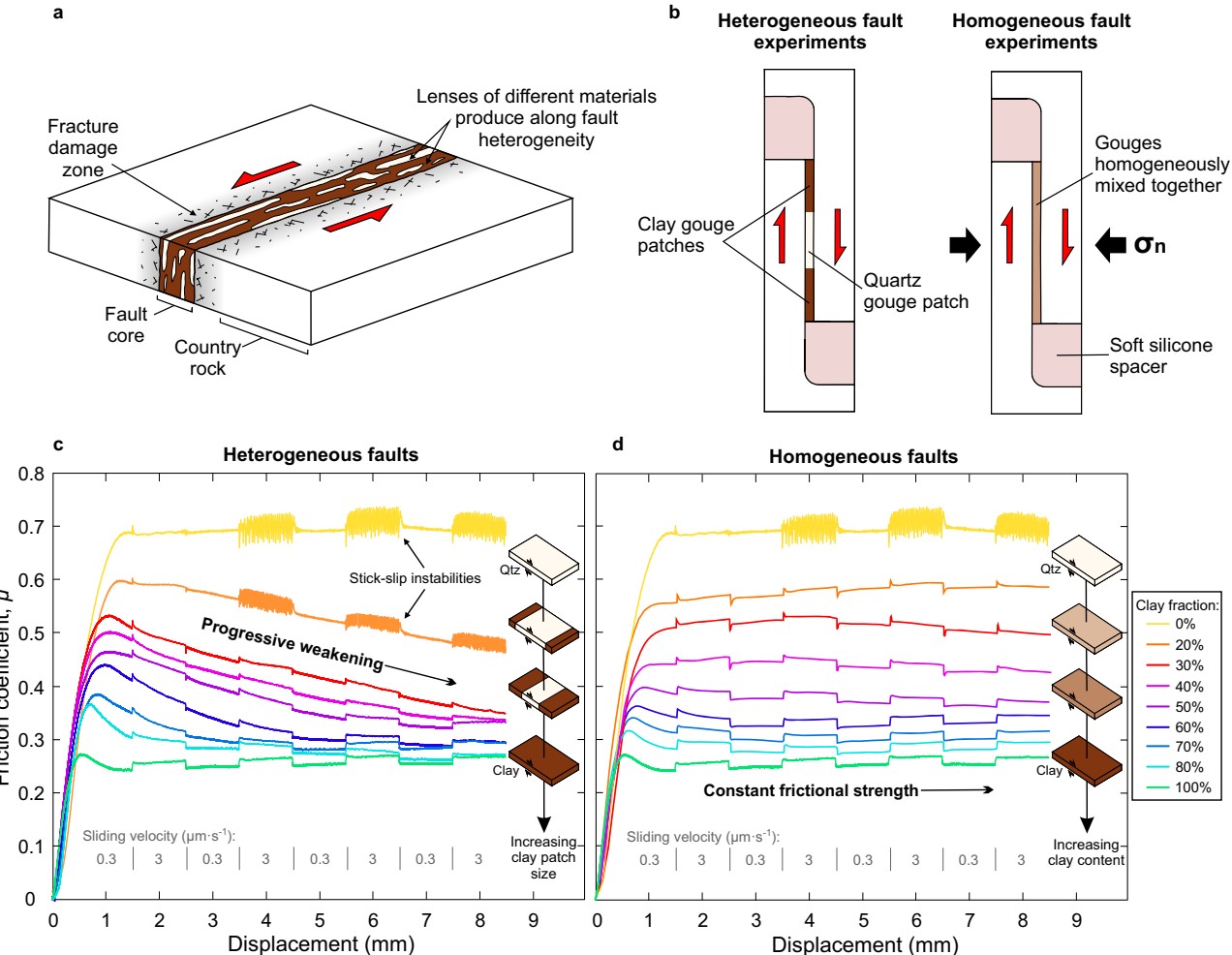

**Fig. 1 Mechanical behaviour of laterally heterogeneous vs. homogeneously mixed clay-quartz fault gouge layers. a** Schematic diagram of a typical natural fault zone showing how lenses of different materials trapped within the fault core produces a heterogeneous structure. **b** Simplified diagrams of the experimental setup for the heterogeneous fault experiments, where quartz and clay gouges are separated into adjacent patches, and homogeneous fault experiments where the two gouges are homogeneously mixed together. **c** Evolution of the friction coefficient ($\mu$) with displacement for the heterogeneous experimental faults and, (**d**) the homogeneous experimental faults. Heterogeneous faults show ubiquitous post-yield weakening with increasing displacement, in contrast to homogeneous faults where $\mu$ remains relatively constant once the layer has yielded after approximately 1 mm of slip. For pure quartz and heterogeneous faults where the quartz patch comprises ≥80% of the total fault area, stick-slip instabilities occur, triggered by up-steps in the sliding velocity.

quartz gouge experiences a greater layer thickness reduction of about 20 μm than the clay gouge during slip (Supplementary Fig. 2). In the heterogeneous layer experiments this would result in an increase of normal stress on the weaker clay patches leading to a progressive reduction in shear resistance, as observed in our experiments. The magnitude of this effect is dependent on the bulk ($K$) and shear ($G$) moduli[35], which are poorly constrained for the gouge materials in this study. Using plausible values for the moduli (Supplementary Note 1) indicates that this differential compaction effect could potentially explain a large component of the weakening we observe in our experiments (Fig. 2d).

**Frictional stability**. The velocity steps from Fig. 1c, d are used to calculate the evolution in the rate-and-state friction[36–38] parameter ($a—b$), which determines the frictional stability of the fault[39–41]. When ($a–b$) > 0, the sliding behaviour is rate-strengthening, suppressing instabilities and promoting stable sliding, whereas when ($a–b$) < 0, the sliding behaviour is rate-weakening which promotes unstable slip behaviour and the occurrence of stick-slips in the laboratory. The values of ($a–b$) are

consistently lower (i.e. less rate-strengthening) in the heterogeneous faults throughout the experiments (Fig. 3a–c). This finding indicates that the heterogeneous faults are closer to the potentially unstable, rate-weakening regime than their homogeneous counterparts.

For the homogeneous faults with the pure quartz gouge and the heterogeneous faults where the quartz patch comprises ≥80% of the total sliding area, only the first velocity step can be used to determine the rate dependence due to the occurrence of stick-slip instabilities triggered by subsequent velocity steps. However, this initial velocity step at 1.5 mm displacement does show negative values of ($a–b$) associated with rate-weakening behaviour (Fig. 3a), which is consistent with the occurrence of stick-slip instabilities later in the experiment. All of the calculated rate-and state friction data are presented in Supplementary Table 1.

## Discussion
Our experiments show that laterally heterogeneous fault gouge layers weaken significantly in comparison to homogeneous layers, pointing to heterogeneity-induced weakening effects. We

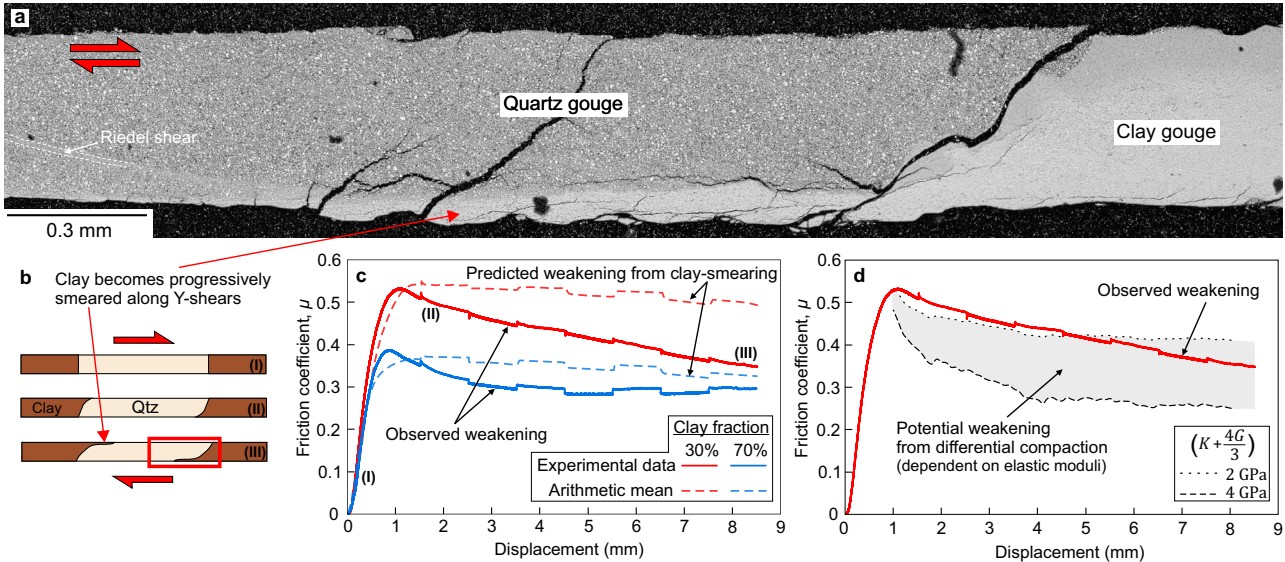

**Fig. 2 Microstructural evolution and potential causes of weakening in the heterogeneous fault gouge layers. a** Backscatter electron image of the interface between a clay-quartz patch recovered at the end of an experiment. The clay phase becomes smeared along a boundary Y-shear plane that propagates into the quartz patch. Since it is difficult to keep the gouge layer intact upon removal from the direct-shear assembly at the end of the experiment, the full extent of the localized shear band was not recovered. **b** Schematic diagram showing the evolution of the fault gouge layers with progressive smearing of the clay phase along localized Y-boundary shears (red box shows the location of the micrograph in (**a**)). **c** Observed weakening versus predicted weakening due to clay smearing for heterogeneous layers comprised of 30% and 70% clay fractions. The predicted weakening is calculated using the arithmetic mean of the friction coefficients of the endmember quartz and clay gouges and by assuming that the length of the clay patches increases by the amount of displacement on the fault as clay is smeared along localized Y-shear planes. The observed weakening is considerably greater than the predicted weakening. The labels (I), (II) and (III) correspond to the structural evolution in (**b**). **d** The potential weakening effect from differential compaction between the clay and quartz gouge patches. This effect is dependent on the bulk (*K*) and shear (*G*) moduli of the gouge, which are poorly constrained (see Supplementary Note 1 for full discussion). The differential compaction could account for a large component of the weakening in the heterogeneous fault experiments.

hypothesize that the weakening occurs due to a combination of mechanisms, all of which can affect natural faults. The mechanical smearing of the weak phase with slip can reduce the overall shear resistance as shear is likely to localize within the weak phase[34], although this mechanism by itself can explain only part of the observed weakening (Fig. 2c, see also Supplementary Fig. 4). Another contributing mechanism can be differential compaction of the weak and strong phases during shear (Supplementary Fig. 2) which would result in a redistribution of normal stress along the shearing layer, with the weaker phase supporting higher normal stresses (see Supplementary Note 1 for further discussion of this effect). The differential compaction can produce significant weakening effects (Fig. 2d) but it is poorly constrained, with the conclusions based on end-member tests of pure quartz and clay samples under constant normal stress, highlighting the need to better capture and characterise the compaction/dilation effects in gouge experiments. Finally, additional weakening can be due to shear occurring in the weaker clay gouge that produces stress concentrations along localized Y-shear bands that propagate through the stronger quartz patches leading to enhanced weakening. Similar shear stress concentrations have also been suggested to promote slip events in strong, rate-weakening gouge patches in recent low normal stress experiments on decimeter-scale heterogeneous faults[42]. Due to difficulty keeping the gouge layer intact during recovery at the end of our experiments, we were unable to acquire detailed microstructural images of the tips of the propagating shear bands to look for evidence of shear/damage zones in the quartz patch. We do, however, observe $R_1$ Riedel shears in the quartz patch (Fig. 2a) which may help facilitate weakening by connecting the smeared clay on opposite sides of the layer.

Competency contrasts between strong and weak materials in shear zone mélanges have been suggested previously to be important in controlling the average fault strength and rheology[43], with only a small amount of well-connected weak material needed to reduce fault strength when structural foliations are well developed[9]. In our experiments, if the gouge layers could be taken to greater shear displacements, the clay smearing we observe along the edges of the quartz patch (Fig. 2a) would ultimately form a through-going layer of interconnected weak material after a few centimetres of slip. Previous work has shown that such through-going layers can lead to a reduction in the frictional strength at slow slip velocities[11] and also increase the efficiency of dynamic weakening at seismic slip velocities (1 m/s)[44]. Although weak phase smearing would, to some extent, homogenize the fault in the overall direction of shear, heterogeneity would likely always be prevalent in natural faults, particularly perpendicular to the slip direction and also at scales larger than investigated in this study, as observed in natural fault zones[25,45]. Our results show that the average frictional strength of laterally heterogeneous faults is not just an average of the respective friction properties (Fig. 2c), and that competency contrasts can substantially reduce the fault strength, even when structural foliations are in their infancy and unconnected (Fig. 2a). They also highlight the need to investigate further how different types of fault heterogeneity, including fault-parallel and fault-normal heterogeneity, and its evolution, affect the frictional behaviour of faults.

Contrasting material properties within fault zones have also been suggested to give rise to mixed fault slip behaviour[46] and exert an important control on earthquake rupture dynamics[47,48]. Heterogeneities are also thought to strongly influence the sliding behaviour of other types of frictional interface, such as at the base

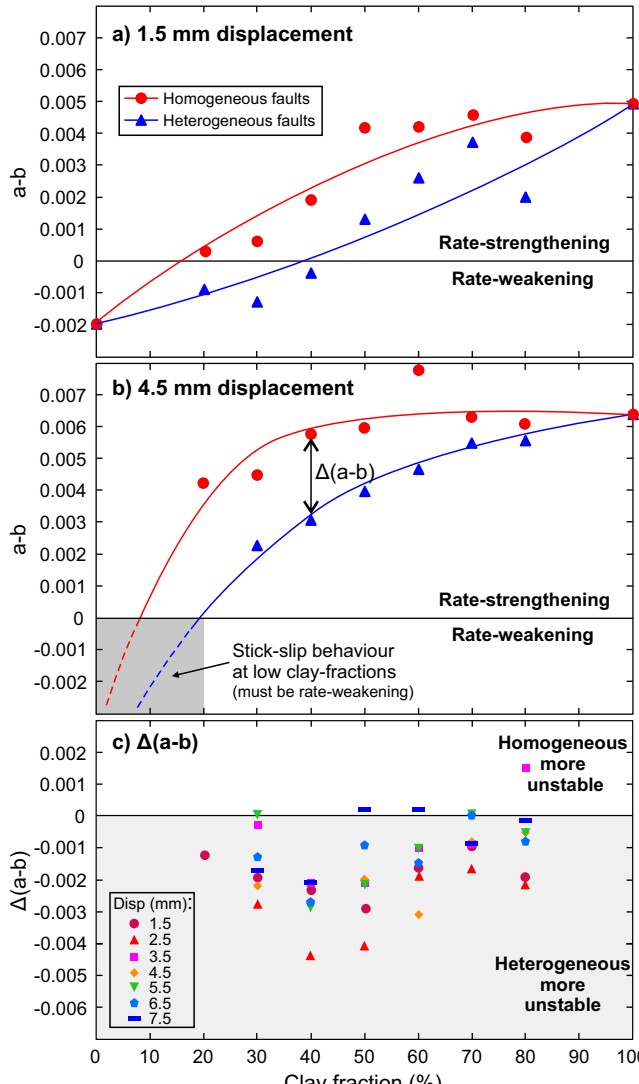

**Fig. 3 Evolution of the stability-controlling rate-and-state friction parameter (a–b) as a function of clay content for the spatially heterogeneous and homogeneously mixed clay-quartz fault gouge layers.** The (a–b) values are compared after (**a**) 1.5 mm and (**b**) 4.5 mm displacements, with the heterogeneous faults having consistently lower values than the homogeneous faults. **c** The difference in (a–b) between the heterogeneous and homogeneous faults, Δ(a–b), is shown for all velocity steps across the entire displacement range, with the majority of values being negative, highlighting that the heterogeneous faults are less stable than their homogeneous counterparts. Note that, for displacements larger than 1.5 mm, the (a–b) values cannot be calculated for the homogeneous quartz fault (i.e., 0% clay fraction) or the heterogeneous fault with a 20% clay fraction, as stick-slip instabilities were triggered by the velocity steps.

of glaciers[49]. Our experiments show that heterogeneity produces an overall reduction in stability when compared to homogeneous faults (Fig. 3). It should be noted that a sufficient amount of rate-weakening material is still required to promote unstable slip. In our experiments, when the proportion of the rate-weakening material is ≤70%, the heterogeneous faults are stable overall, with positive (a–b) values, although the values are closer to zero (and hence rate-neutral behaviour) than those of their homogeneous counterparts (Fig. 3); however, the behaviour remains rate-strengthening, instabilities do not initiate and aseismic slip prevails. Only when the strong rate-weakening patch comprises ≥80% of the layer do stick-slip instabilities occur (Fig. 1c). As

shown previously in experimental studies on rate-weakening quartz gouges, microstructural evolution and deformation localization into discrete shear bands is a prerequisite for unstable stick-slip behaviour[50–52]. Therefore, in the heterogeneous faults, slip behaviour would be dictated by the competing processes of fault stabilization via deformation in weak rate-strengthening materials, versus destabilization caused by localization within the strong rate-weakening patches. When the strong rate-weakening patches are large enough for their internal structure to evolve independently, stick-slip instability may occur.

The role of heterogeneity is summarized in Fig. 4, where, for a given clay-quartz mixture, heterogeneous faults are weaker and less stable relative to their homogeneous equivalents. Although it is often invoked that large-scale heterogeneities are responsible for the spectrum of slip behaviour observed on natural faults[17,18], the results presented here highlight the potential of small-scale heterogeneities, which are also abundant in natural fault zones[9,10,22], to exert a significant control on fault zone strength and stability. There are similarities between the slip behaviour we observe in our small-scale heterogeneous experiments and how large-scale heterogeneities are thought to control the behaviour of natural faults. For example, decreasing the size of the rate-weakening patch makes the response more stable in both our experiments and numerical modelling[53], as can be intuitively expected and consistent with stability studies of rate-and-state faults that slip instability can only result from large enough rate-weakening patches[39]. At the same time, small-scale fault zone heterogeneity would more readily evolve with shear, and hence may depend on the fault maturity, healing processes, and spatio-temporal history of fault slip.

To summarize, we show that, by introducing a simple heterogeneous structure into a fault zone, the fault strength is substantially reduced and the stability of the experimental fault is overall decreased in comparison to compositionally identical but homogeneously mixed gouges. Our data, along with the abundance and complexity of heterogeneity that occurs over many different scales in nature[9,10,22–25], suggest that interactions between heterogeneously distributed materials with different frictional properties likely exerts an important control over the mechanical strength and influences whether tectonic faults experience aseismic or earthquake slip. The smaller the scale of heterogeneity, the more likely it is to be intractable in modelling earthquake source processes and hence ignored. These considerations, together with our findings, necessitate further laboratory experiments and modelling to study the effects and evolution of fault rock heterogeneity within complex fault zones, to enable the quantification and inclusion of the smaller-scale heterogeneity effects into larger-scale constitutive laws for modelling fault processes of societal interest, such as nucleation of natural and induced earthquakes.

## Methods

**Experimental procedure.** The gouge layers are deformed in a direct-shear arrangement (Supplementary Fig. 1) within a triaxial deformation apparatus[54]. The layers (~1.3 mm initial thickness prior to pressurization), prepared in either heterogeneous patches or as a homogeneous quartz-clay mixture, are placed between the direct-shear forcing blocks and soft silicone spacers are positioned at each end so that displacement can be accommodated without supporting any load (Supplementary Fig. 1). To discourage boundary shear at the edges of the gouge layer, the sliding area (50 × 20 mm) on the forcing blocks contains grooves cut perpendicular to the sliding direction (200 μm deep with 400 μm spacing). Once the gouge layer is constructed, the direct-shear arrangement is surrounded by a low-friction polytetrafluoroethylene (PTFE) sleeve (0.25 mm thickness) to minimize jacket friction in the vicinity of the layer, before being placed into a soft, 3 mm thick, PVC jacket (Nalgene 180 clear tubing). The jacketed direct-shear arrangement is then placed in between the platens of the sample assembly which is inserted into the pressure vessel of the triaxial apparatus. In this geometry, the normal stress ($\sigma_n$) is applied to the gouge layer by the confining pressure. The pore-fluid pressure is introduced to the layer through three porous disks, embedded in each direct-shear

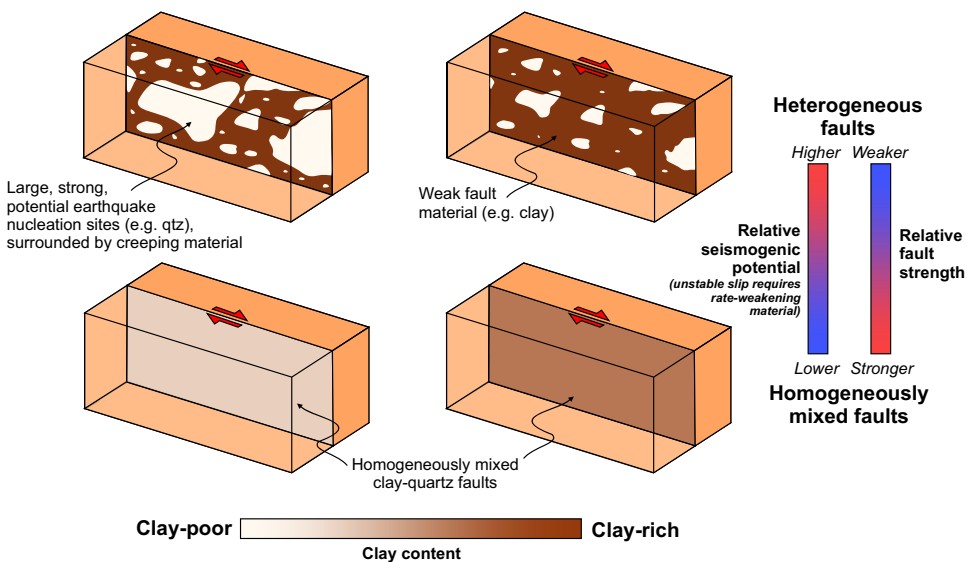

**Fig. 4 Schematic fault model showing the effect of heterogeneity on fault strength and stability.** For a given clay-quartz composition, the introduction of heterogeneity (i.e. the separation of clay and quartz into patches) leads to a reduction in fault strength relative to a homogeneously equivalent fault and also a decrease in stability, increasing the likelihood of seismogenesis on the fault.

forcing block, which are positioned to ensure an even distribution of pore fluid throughout the layer. Deionized water is used as the pore fluid. Both the confining and pore-fluid pressures are held constant throughout the experiments by servo-controlled pumps on each pressure system, with a resolution better than 0.01 MPa. Linear variable differential transformers (LVDTs) are attached to the pistons of the servo-control pumps, meaning that the volume of fluid expelled from the sample as it compacts during shearing can be monitored as the pressure is held constant. We therefore use the pore pressure pump as a pore volumometer to track the evolution of layer thickness during our experiments (Supplementary Fig. 2); we assume that the sliding area remains constant and that all volumetric strain is accommodated by a change in layer thickness. The gouge layers are sheared by the axial piston of the triaxial apparatus and velocity steps are imposed to calculate the rate-and-state friction parameters. The evolution of shear stress is monitored by an internal force gauge within the axial piston, with a measurement resolution of better than 0.05 kN.

## Data availability

The associated experimental data files for this research can be accessed in National Geoscience Data Center (NGDC) via the following link: https://webapps.bgs.ac.uk/services/ngdc/accessions/index.html#item164865.

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

## Acknowledgements

Gary Coughlan is thanked for assistance in developing and maintaining the experimental apparatus. We are grateful to Elisabetta Mariani for help with and maintenance of the SEM facilities. This work is supported by Natural Environment Research Council grant NE/P002943/1.

## Author contributions

J.D.B., D.R.F. and N.L. developed the main ideas. J.D.B. performed the experiments, ran microstructural analyses and produced the initial manuscript. All authors contributed to interpreting the results and editing the manuscript.

## Competing interests

The authors declare no competing interests.
