## [Peer Review File · Nature Communications]

REVIEWER COMMENTS

Reviewer #1 (Remarks to the Author):

In this interesting manuscript, by means of newly-conceived experiments, it is shown that small-scale mineral heterogeneity in fault slip zones controls both fault strength (or the maximum stress at which a fault fails producing earthquakes or aseismic creep) and the response of the fault to perturbations in the loading conditions (e.g., far field velocity, stress). This latter response of the fault is relevant: if, during slip, the fault weakens (or the dynamic friction decreases) faster than the rate at which the elastic strain energy stored in the fault wall rocks is released, this will result in a frictional instability (= earthquake ruptures); if the dynamic friction increases or, alternatively, decreases slower than the rate at which the elastic strain energy is released, this will result in aseismic slow slip transients).

But what is "fault rock heterogeneity"? Typically, laboratory experiments are conducted on (1) synthetic fault gouges made by one mineral, (2) synthetic fault gouges made by mixtures of two or more minerals and (3) natural fault gouges retrieved from outcrops or fault drilling projects (almost all natural gouges consist of an assemblage of grains with different composition: clay, quartz, feldspar, calcite, dolomite, etc.). In cases (2) and (3) the experimental fault is heterogeneous in composition (in this manuscript, this mineral assemblage is called "homogeneously mixed" fault gouge). There are several thousands of published data about friction experiments conducted on these mixtures. Instead, there are very few laboratory studies which reproduce natural slip zones where "fault rock heterogeneity" is related to the spatial arrangement of two or more minerals in the experimental fault. For instance, motivated by the spatial arrangement of different minerals found in natural slip zones of active seismogenic faults, Smeraglia et al. (Scientific Reports, 2017) considered the "fault rock heterogeneity" as sub-parallel to the fault slip surface. In this case, heterogeneity consists in a compositional layering with clays next to the slip surface and calcite grains beneath.

The novelty of this study submitted to Nature Communication is that, for the first time to my knowledge, the "fault rock heterogeneity" (i.e., layers of clays alternated to quartz-built gouges) is perpendicular to the slip surface (Fig. 1b). At the millimetre to sub-millimetre scale such spatial organization can be found in faults which cut across foliated rocks (e.g., quartzites, micaschists). For instance, in the case of impure greenschists facies quartzites, mm-cm thick layers made of quartz grains are alternated to sub-millimetre-thick layers made of chlorite and white micas.

Thanks to this newly conceived experimental configuration, the authors show that the small scale spatial mineralogical heterogeneity (layering) has profound effects on the frictional behaviour of the experimental fault. In particular, (1) the friction coefficient of the layered heterogeneous gouges decreases with increasing fault slip (this slip weakening behaviour is not observed in experiments performed on "homogeneously mixed" fault gouges, Fig. 1c-d) and (2), layered heterogeneous fault gouges are, regarding the response of the friction coefficient to loading perturbations, more unstable; that is, they are more prone to trigger frictional instabilities and laboratory earthquakes than "homogeneously mixed" fault gouges (Fig. 3).

I find the description of the experimental configuration and of the results well-written and the conclusions based on solid experimental evidence. The figures are well-designed. I also suggest to keep the supplemental material as it is (i.e., I really enjoyed the discussion about the effect of differential compaction on interface weakening). In conclusion, the manuscript is nicely written and surely deserves to be published in Nature Communications after considering or discussing the following minor point:

My field and experimental experience is that at the scale of natural slip zones (< 2 cm thick, e.g., Sibson 2003, Bulletin of the Seismological Society of America) the compositional heterogeneity is parallel to the fault slip surfaces (e.g., clay-rich layers sub-parallel to calcite-rich layer in the case of exhumed seismogenic faults in the Central Apennines, see Smeraglia et al., Scientific Reports 2017) or, very common as the authors of this manuscript well know, that the mineral composition in the slip zone is a mixture of minerals (homogeneously mixed fault gouges). This spatial or mineralogical heterogeneity is easily explained by the smearing and dragging of the wall rock materials and their mixing in the slip zone, as also shown in Fig. 2a of this manuscript, or by other processes occurring in natural faults (incongruent pressure-solution-precipitation, neo-formation of clay minerals in natural fault cores, etc.).

In this manuscript is discussed a "fault rock heterogeneity" (i.e., layers of clays alternated to

quartz-built gouges) perpendicular to the fault slip surface (Fig. 1b). However, as explained above, such initial "micro-scale" spatial heterogeneity or compositional layering is lost during initial shearing of the gouges (Fig. 2a). As a consequence, I suggest the authors to reconsider (or discuss by introducing some natural examples) the last statement of the abstract (lines 20-24):

"The results demonstrate that small-scale geological heterogeneity has pronounced effects on fault strength and stability, and by extension on the occurrence of slow-slip transients versus earthquake ruptures and the characteristics of the resulting events, and should be incorporated in lab experiments, fault friction laws, and earthquake source modeling."

In fact, the mechanical data of the experiments discussed in this manuscript are representative of the very first mm or cm of slip (or slip initiation, depending on the thickness of the layered rocks) in natural faults cutting layered rocks. Such small-scale heterogeneity perpendicular to the fault slip surface will be lost after few millimetres or tens of centimetres at most of slip in natural faults and, as a consequence, these experiments cannot reproduce the conditions that lead to slow-slip transients versus earthquake ruptures in large displacement faults. Instead, if the authors consider their experimental configuration as an analogue, to be scaled, of large-scale spatial heterogeneity (see their figure 4) associated to large displacement faults which put in contact rocks with different mineral composition and mechanical properties, I do agree with their conclusion.

Congratulations to the authors for this nice piece of work,

Giulio Di Toro

Reviewer #2 (Remarks to the Author):

This is a really interesting paper and I think it should be published in Nature Communications, but I have some reservations that would need to be addressed.

The topic is very appropriate for Nature Communications. This paper addresses interesting questions that are topical and cutting-edge in earthquake physics: what causes unstable frictional sliding and what causes the spectrum of slip behaviors on tectonic faults. The experiments are elegant, the results are clear and the lab work is very well described. But my sense is that some of the interpretation needs to be reoriented.

My main concern is that I'm not sure I agree with the title. Does heterogeneity really promote unstable slip? The experiments shown indicate that stability is dictated mainly by the clay/qtz fraction, right? I appreciate the results in Fig 3 but it seems like the main effect there is simply the result of incomplete mixing given the available shear displacement. The data of Fig 3c show that the effect decreases with shear displacement. Doesn't this imply that the main factor causing a reduction in a-b (as seen in Fig3a/b) is the role of the quartz patch? The quartz is velocity weakening so the fault has more of a tendency for velocity weakening when the fault has a separate quartz patch. As the quartz patch becomes increasingly mixed the homogeneous and heterogeneous faults have similar behavior. So, I'm not sure I agree with the idea that heterogeneity is the key ingredient. Another way to make this point is: if heterogeneity itself promotes unstable slip shouldn't I expect that to happen for a heterogeneous mixture of weak, velocity strengthening materials?

A related point involves the data of Figure 3a. My sense is that these are a key part of the story but when I look at the raw data of Figure 1c (or supplementary Fig 3) I only see velocity weakening for clay fractions of 0 and 20%. The raw data for 30 and 40% clay are velocity strengthening at all displacements and the a-b values for the velocity step at 1.5 mm would be even larger if the strain weakening trend from 1-1.5 mm were accounted for. This doesn't negate the importance of Fig 3a but it reduces it. In any case it's important to make sure readers can follow your analysis.

Minor points in no particular order

1. How do you measure layer thickness during shear? Hopefully this could be added to the supplement. The overall thinning seems small; the layer goes from 1 mm to 0.85 mm after a shear strain of 10?
2. How do you discourage or eliminate boundary shear at the edge of the forcing block?
3. Please provide a bit more detail on the jacket. PVC can be used for rigid plumbing pipe. I assume you're using another type and it would be good to provide this information in the supplement.
4. Your references are very good. You might also consider commenting on the comparison of your work with that of Niemeijer et al. GRL 2010 (Fabric induced weakness of tectonic faults).

5. A useful addition to your discussion of Figure 4 would be to connect spatial heterogeneity to the slip patch size for rupture nucleation/propagation. One way to produce slow slip is to limit the size of the velocity weakening region (for example as shown by Liu & Rice 2005/2007) and your data provide some nice connections to these ideas.

6. I like the idea of the color bars for strength and seismic potential in Fig. 4 but I'm not sure I see the argument for low strength and high seismogenic potential. Maybe this is just the same point I raised above, but here you seem to be going beyond the idea of a relative change (heterogeneity promotes unstable slip) to a more definite statement.

7. It would be useful to comment on how the kind of heterogeneity you envision would be sustained in a tectonic fault zone with sizable offset. For example, one could imagine that a fault zone with initial heterogeneity of the sort you examine would become homogeneously mixed after a relatively small offset.

Chris Marone

Reviewer #3 (Remarks to the Author):

Review of 315765-0, "Fault rock heterogeneity produces fault weakness and promotes unstable slip" by Bedford, Faulkner and Lapsuta.
By Terry E. Tullis

General Comments

My comments are both for the editor and the authors. This is a good paper and could be published as is. It is clear and succinctly written and the noteworthy results are made clear in the paper, namely that heterogeneous distribution of rock types along a fault contribute to the strength and stability of the fault in non-intuitive ways and that this is important in modeling earthquakes. It is a significant contribution to the field. The work and methodology is sound and supports the conclusions. Enough detail is presented that the results could be reproduced. However, I have some suggestions for the authors to consider that should not involve much additional work and should improve an already fine paper. I have included these as comments in both the Word file for the article and the Word file for the supplement, using Word's Comments feature under Word's Review tab.

I have copied and pasted below all my marginal comments, but they are better understood in the context of where they appear in the Word doc. Although there are a number of comments scattered among my marginal comments, perhaps the very last one on the Supplement file is the most important, as it affects the reader's understanding of what the authors conclude about the correct explanation for their experimental observations.

Comments in the Article file:

Line 28: μ should be τ

Line 129-130: However, if the clay is supporting more of the normal stress and therefore is contributing more than its volumetric proportion to the overall measured properties, why is the velocity dependence seem to be skewed toward the rate weakening quartz? Have you tried to do a quantitative model of a-b as you do for μ ? Of course it's not clear how one should do that - I suppose just treat a-b as a parameter with values that go with each phase and that sum up linearly in proportion to their contact area along the two bounding faults as you do for the friction. Not sure if that makes sense or not!

Line 135-136: Have you seen any evidence of this in thin sections? Seems as if there might be shear/damage zones in the quartz connecting the tips of the smeared clay on opposite sides of the overall layer. In any case, some comment as to whether this has been observed directly would be helpful as it seems a likely process. Related to this, see my note commenting on Figure 2.

Line 174-177: This overall problem sort of reminds me of the complexities of deformation of a poly-phase aggregate, where neither the uniform stress or uniform strain models adequately describe the behaviour and some more detailed modelling is needed. Your experimental geometry nominally simpler than that, but even so the simple model predictions don't seem to adequately explain the results. With patches on a fault plane rather than simple strips perpendicular to the slip direction as in the experiments, the situation will clearly be even more complex.

Line 343-344: I'm thinking that it would be helpful to show calculated curves for 60, 70, and 80 percent clay. Given that the length of the quartz patch along the fault is progressively smaller in

those cases, the decline of the calculated curves as a function of slip would occur sooner and sooner as the clay content goes up. This is seen in the data. So looking at this might help to better understand the proportion of the weakening due to clay smearing. This, combined with localized stress concentrations as you discuss on lines 135-136, could help explain the observations.

Comments in the Supplement file:

Caption to Supplementary Figure 1: It would help if you give the length of the sample so one is able to think about how the amount of displacement as the experiments progress compares with that length. Without that information one is unable to reproduce your calculations in Figure 2c. Perhaps your figure is to scale and the 20 mm diam. platens is all you need, but the figure could be more schematic.

"Similar weakening" in caption for Supplementary Figure 3: I note however that the value of $a-b$ seems to be more positive than in the configuration with the quartz patch in the middle and in fact is similar to the homogeneous gouge (Figure 1d). I presume you have no explanation for this difference, assuming it is significant, which appears to be the case. Either say that you have no explanation for it or maybe you can come up with one. One value of making a comment to this effect is that when some reader notices it you will not seem to be unaware of it! I suggest you include the data for these experiments in Table S1.

First line of very last paragraph: In spite of these caveats I think it is worth considering an attempt to add another component to Figure 2, i.e. Figure 2d, that includes an attempt to calculate the curves for one or more cases of clay fraction as you do in Figure 2c. I assume you could take the data at each displacement and run through the calculation as a function of displacement. It is always advisable for your figures to tell your story since that is what many readers will focus on. As the paper now stands, Figure 2c leads one to conclude that you don't really understand the extra observed weakening. Reading the lines near the end of the previous paragraph in the supplement is needed to realize that you actually do have a viable explanation for the weakening. This location hides that understanding too much! The new figure could, as does part c now, only focus on each effect by itself, leaving to words the statement that one looks larger than the other and together they could both clearly do the job.

Response to reviewers' comments on "Fault rock heterogeneity produces fault weakness and reduces fault stability"

In this document, we outline our response to the reviewers' comments on the original manuscript. We would like to sincerely thank all three reviewers for their constructive comments; the manuscript has been significantly improved by their suggestions. In part, we have added additional text into the discussion of the manuscript to clarify how our results relate to heterogeneity observed in natural fault zones and to emphasize the importance of further study of fault heterogeneity and its evolution. We have modified the title and added extra discussion on the role of heterogeneity in modifying stability and the requirement of having sufficient rate-weakening material in the fault. We have also added an additional panel into Fig. 2 to show the potential magnitude of the weakening effect from differential compaction and a supplementary Figure (Fig. S4) to illustrate that the effects of the differential compaction are compatible with the friction stability changes. Our main results and conclusions remain the same.

Please find below our detailed point-by-point response to the reviewers' comments on the original manuscript. Our responses are written in black font and any line numbers to which we refer to are for the new redlined version of the manuscript where our changes are documented.

Blue: reviewer comment.

Black: authors' response.

Reviewer 1 comments

In this interesting manuscript, by means of newly-conceived experiments, it is shown that small-scale mineral heterogeneity in fault slip zones controls both fault strength (or the maximum stress at which a fault fails producing earthquakes or aseismic creep) and the response of the fault to perturbations in the loading conditions (e.g., far field velocity, stress). This latter response of the fault is relevant: if, during slip, the fault weakens (or the dynamic friction decreases) faster than the rate at which the elastic strain energy stored in the fault wall rocks is released, this will result in a frictional instability (= earthquake ruptures); if the dynamic friction increases or, alternatively, decreases slower than the rate at which the elastic strain energy is released, this will result in aseismic slow slip transients).

But what is “fault rock heterogeneity”? Typically, laboratory experiments are conducted on (1) synthetic fault gouges made by one mineral, (2) synthetic fault gouges made by mixtures of two or more minerals and (3) natural fault gouges retrieved from outcrops or fault drilling projects (almost all natural gouges consist of an assemblage of grains with different composition: clay, quartz, feldspar, calcite, dolomite, etc.). In cases (2) and (3) the experimental fault is heterogeneous in composition (in this manuscript, this mineral assemblage is called “homogeneously mixed” fault gouge). There are several thousands of published data about friction experiments conducted on these mixtures. Instead, there are very few laboratory studies which reproduce natural slip zones where “fault rock heterogeneity” is related to the spatial arrangement of two or more minerals in the experimental fault. For instance, motivated by the spatial arrangement of different minerals found in natural slip zones of active seismogenic faults, Smeraglia et al. (Scientific Reports, 2017) considered the “fault rock heterogeneity” as sub-parallel to the fault slip surface. In this case, heterogeneity consists in a compositional layering with clays next to the slip surface and calcite grains beneath.

The novelty of this study submitted to Nature Communication is that, for the first time to my knowledge, the “fault rock heterogeneity” (i.e., layers of clays alternated to quartz-built gouges) is perpendicular to the slip surface (Fig. 1b). At the millimetre to sub-millimetre scale such spatial organization can be found in faults which cut across foliated rocks (e.g., quartzites, micaschists). For instance, in the case of impure greenschists facies quartzites, mm-cm thick layers made of quartz grains are alternated to sub-millimetre-thick layers made of chlorite and white micas.

Thanks to this newly conceived experimental configuration, the authors show that the small scale spatial mineralogical heterogeneity (layering) has profound effects on the frictional behaviour of the experimental fault. In particular, (1) the friction coefficient of the layered heterogeneous gouges decreases with increasing fault slip (this slip weakening behaviour is not observed in experiments performed on “homogeneously mixed” fault gouges, Fig. 1c-d) and (2), layered heterogeneous fault gouges are, regarding the response of the friction coefficient to loading perturbations, more unstable; that is, they are more prone to trigger frictional instabilities and laboratory earthquakes than “homogeneously mixed” fault gouges (Fig. 3).

I find the description of the experimental configuration and of the results well-written and the conclusions based on solid experimental evidence. The figures are well-designed. I also suggest to keep the supplemental material as it is (i.e., I really enjoyed the discussion about the effect of differential compaction on interface weakening). In conclusion, the manuscript is nicely written and surely deserves to be published in Nature Communications after considering or discussing the following minor point:

We are grateful to the reviewer for the positive comments on our manuscript, and also for detailing the importance of our work and how it fits into the bigger picture of fault mechanics. We also thank the reviewer for bringing the reference of Smeraglia et al., (2017) to our attention; it is very relevant to our work and we have included it in the manuscript (ref. number 44).

My field and experimental experience is that at the scale of natural slip zones (< 2 cm thick, e.g., Sibson 2003, Bulletin of the Seismological Society of America) the compositional heterogeneity is parallel to the fault slip surfaces (e.g., clay-rich layers sub-parallel to calcite-rich layer in the case of exhumed seismogenic faults in the Central Apennines, see Smeraglia et al., Scientific Reports 2017) or, very common as the authors of this manuscript well know, that the mineral composition in the slip zone is a mixture of minerals (homogeneously mixed fault gouges). This spatial or mineralogical heterogeneity is easily explained by the smearing and dragging of the wall rock materials and their mixing in the slip zone, as also shown in Fig. 2a of this manuscript, or by other processes occurring in natural faults (incongruent pressure-solution-precipitation, neo-formation of clay minerals in natural fault cores, etc.).

In this manuscript is discussed a “fault rock heterogeneity” (i.e., layers of clays alternated to quartz-built gouges) perpendicular to the fault slip surface (Fig. 1b). However, as explained above, such initial “micro-scale” spatial heterogeneity or compositional layering is lost during initial shearing of the gouges (Fig. 2a). As a consequence, I suggest the authors to reconsider (or discuss by introducing some natural examples) the last statement of the abstract (lines 20-24):

“The results demonstrate that small-scale geological heterogeneity has pronounced effects on fault strength and stability, and by extension on the occurrence of slow-slip transients versus earthquake ruptures and the characteristics of the resulting events, and should be incorporated in lab experiments, fault friction laws, and earthquake source modeling.”

In fact, the mechanical data of the experiments discussed in this manuscript are representative of the very first mm or cm of slip (or slip initiation, depending on the thickness of the layered rocks) in natural faults cutting layered rocks. Such small-scale heterogeneity perpendicular to the fault slip surface will be lost after few millimetres or tens of centimetres at most of slip in natural faults and, as a consequence, these experiments cannot reproduce the conditions that lead to slow-slip transients versus earthquake ruptures in large displacement faults. Instead, if the authors consider their experimental configuration as an analogue, to be scaled, of large-scale spatial heterogeneity (see their figure 4) associated to large displacement faults which put in contact rocks with different mineral composition and mechanical properties, I do agree with their conclusion.

Congratulations to the authors for this nice piece of work,
Giulio Di Toro

We agree that, in our simple experimental setup, if the layer could be taken to greater shear displacement, clay smearing would encapsulate the quartz patch after a few centimeters of shear and thus the heterogeneity would essentially be lost. However, this view only considers the loss of heterogeneity parallel to the overall, large-scale, slip direction. In a non-planar fault zone with a complex internal structure, slip may also locally occur in directions different from the slip vector, allowing heterogeneity to potentially persist even for large displacement. Hence, in natural fault zones, we would expect heterogeneity to persist over different scales. While some fault-zone studies indeed find localized through-going layers that are relatively homogeneously mixed, as discussed by the reviewer,

complex compositional heterogeneity is still commonly observed in natural fault zones, such as the Carboneras Fault (e.g., Figure 5a of Rutter et al., 2012, JSG) and the Punchbowl Fault where there is heterogeneity on either side of the principal slip surface from both the different types of ultracataclastite and the different host rocks (e.g., Fig. 4 of Chester and Chester, 1998, Tectonophysics).

One of the main outcomes of our study is that it demonstrates that fault strength is not just an average of the respective frictional properties of the different patches (Fig. 2c), but rather there are a series of additional weakening effects that are caused by having a spatially heterogeneous distribution of fault materials. Of course, more work is required to document how different types of spatial heterogeneity affect the mechanical behaviour of faults (e.g., Niemeijer et al., 2010). However, we think that our statement at the end of the abstract is still valid, overall. At the same time, to highlight this discussion and complexity of the issue in the manuscript, we have slightly modified the abstract and have added additional text to the manuscript.

The end of the abstract now reads (changes italicized):

“The results demonstrate that geological heterogeneity and its evolution can have pronounced effects on fault strength and stability and, by extension, on the occurrence of slow-slip transients versus earthquake ruptures and the characteristics of the resulting events, and should be further studied in lab experiments and earthquake source modelling.”

On lines 160-173, we now discuss how heterogeneity might be altered and preserved in natural fault zones:

In our experiments, if the gouge layers could be taken to greater shear displacements, the clay smearing we observe along the edges of the quartz patch (Fig. 2a) would ultimately form a through-going layer of interconnected weak material after a few centimetres of slip. Previous work has shown that such through-going layers can lead to a reduction in the frictional strength at slow slip velocities¹¹ and also increase the efficiency of dynamic weakening at seismic slip velocities (1 m/s)⁴⁴. Although weak phase smearing would, to some extent, homogenize the fault in the overall direction of shear, heterogeneity would likely always be prevalent in natural faults, particularly perpendicular to the slip direction and also at scales larger than investigated in this study, as observed in natural fault zones^{25,45}. Our results show that the average frictional strength of laterally heterogeneous faults is not just an average of the respective friction properties (Fig. 2c), and that competency contrasts can substantially reduce the fault strength, even when structural foliations are in their infancy and unconnected (Fig. 2a). They also highlight the need to investigate further how different types of fault heterogeneity, including fault-parallel and fault-normal heterogeneity, and its evolution, affect the frictional behaviour of faults.

Following the suggestion of the reviewer, we have included, on lines 200-207, a mention of the potential analogy between our experiments and effects of large-scale heterogeneity; we have also highlighted that the small-scale heterogeneity can more easily evolve:

There are similarities between the slip behaviour we observe in our small-scale heterogeneous experiments and how large-scale heterogeneities are thought to control the behaviour of natural faults. For example, decreasing the size of the rate-weakening patch makes the response more stable in both our experiments and numerical modelling⁵³, as can be intuitively expected and consistent with stability studies of rate-and-state faults that slip instability can only result from large enough rate-weakening patches³⁹. At the same time, small-scale fault zone heterogeneity would more readily evolve with shear, and hence may depend on the fault maturity, healing processes, and spatio-temporal history of fault slip.

Finally, we modified the end of the manuscript to state (changes italicized):

“These considerations, together with our findings, necessitates further laboratory experiments and modelling to study the effects *and evolution of fault rock heterogeneity within complex fault zones*, to enable *the quantification and* inclusion of the small-scale heterogeneity effects into larger-scale constitutive laws for modelling fault processes of societal interest, such as nucleation of natural and induced earthquakes.”

Reviewer 2 comments

This is a really interesting paper and I think it should be published in Nature Communications, but I have some reservations that would need to be addressed.

The topic is very appropriate for Nature Communications. This paper addresses interesting questions that are topical and cutting-edge in earthquake physics: what causes unstable frictional sliding and what causes the spectrum of slip behaviors on tectonic faults. The experiments are elegant, the results are clear and the lab work is very well described. But my sense is that some of the interpretation needs to be reoriented.

My main concern is that I'm not sure I agree with the title. Does heterogeneity really promote unstable slip? The experiments shown indicate that stability is dictated mainly by the clay/qtz fraction, right? I appreciate the results in Fig 3 but it seems like the main effect there is simply the result of incomplete mixing given the available shear displacement. The data of Fig 3c show that the effect decreases with shear displacement. Doesn't this imply that the main factor causing a reduction in a-b (as seen in Fig3a/b) is the role of the quartz patch? The quartz is velocity weakening so the fault has more of a tendency for velocity weakening when the fault has a separate quartz patch. As the quartz patch becomes increasingly mixed the homogeneous and heterogeneous faults have similar behavior. So, I'm not sure I agree with the idea that heterogeneity is the key ingredient. Another way to make this point is: if heterogeneity itself promotes unstable slip shouldn't I expect that to happen for a heterogeneous mixture of weak, velocity strengthening materials?

A related point involves the data of Figure 3a. My sense is that these are a key part of the story but when I look at the raw data of Figure 1c (or supplementary Fig 3) I only see velocity weakening for clay fractions of 0 and 20%. The raw data for 30 and 40% clay are velocity strengthening at all displacements and the a-b values for the velocity step at 1.5 mm would be even larger if the strain weakening trend from 1-1.5 mm were accounted for. This doesn't negate the importance of Fig 3a but it reduces it. In any

case it's important to make sure readers can follow your analysis.

We thank the reviewer for raising this important point. Indeed, heterogeneity may not always promote instability or reduce stability, as the reviewer's example of mixed rate-strengthening gouges indicates. Similarly, heterogeneity may not always result in fault weakness. Hence we have modified the title to state ***"Fault rock heterogeneity can produce fault weakness and reduce fault stability."*** Further, we agree that unstable behaviour requires a significant fraction of rate-weakening material (e.g., quartz) to be present in the fault, regardless of whether it has a heterogeneous structure or not. We have added additional text to the discussion section of the manuscript on lines 177-187 to emphasize this point in more detail:

Our experiments show that heterogeneity produces an overall reduction in stability when compared to homogeneous faults (Fig. 3). It should be noted that a sufficient amount of rate-weakening material is still required to promote unstable slip. In our experiments, when the proportion of the rate-weakening material is $\leq 70\%$, the heterogeneous faults are stable overall, with positive $(a - b)$ values, although the values are closer to zero (and hence rate-neutral behaviour) than those of their homogeneous counterparts (Fig. 3); however the behaviour remains rate-strengthening, instabilities do not initiate and aseismic slip prevails. Only when the strong rate-weakening patch comprises $\geq 80\%$ of the layer do stick-slip instabilities occur (Fig. 1c).

Minor points in no particular order

1. How do you measure layer thickness during shear? Hopefully this could be added to the supplement. The overall thinning seems small; the layer goes from 1 mm to 0.85 mm after a shear strain of 10?

We use pore volumetry to track the layer thickness evolution during shear; we assume that the sliding area remains constant and that all volumetric strain is accommodated by a change in layer thickness. The layer thickness values in Supplementary Fig. 2b were calculated by measuring the gouge layer thickness at the end of the experiment using a micrometer and then back-calculating the layer thickness evolution during the experiment using the pore volume data, as was also done by Faulkner et al., (2018). We have now stated in the caption of Supplementary Figure 2 how the layer thickness was measured:

At the end of the experiment, the layer thicknesses were measured using a micrometer to be almost identical. The thickness evolution during the experiment was then back-calculated using the pore volume data, assuming the sliding area remains constant and that all volumetric strain is accommodated by a change in layer thickness³³ (see Methods).

We also now give more detail on the pore volumetry in the methods section (lines 240-245 of the main article):

Linear variable differential transformers (LVDTs) are attached to the pistons of the servo-control pumps, meaning that the volume of fluid expelled from the sample as it compacts during shearing can be monitored as the pressure is held constant. We therefore use the pore pressure pump as a pore volumeter to track the evolution of layer thickness during our experiments (Supplementary Fig. 2); we assume that sliding area remains constant and that all volumetric strain is accommodated by a change in layer thickness.

Most of the layer thinning actually occurs during pressurization of the gouge layer to the starting conditions (effective normal stress of 40 MPa). The layer thickness prior to pressurization is about 1.3 mm, which we now clarify on line 225. The 1 mm initial thickness quoted in the main text is actually the layer thickness at the onset of shear, after initial pressurization, which we now clarify on line 60. The magnitude of layer thinning we observe during our experiments is comparable to layer thinning observed in previous studies using a direct-shear setup (e.g., Faulkner et al., 2018).

2. How do you discourage or eliminate boundary shear at the edge of the forcing block?

To discourage boundary shear at the edges of the gouge layer, the direct-shear forcing blocks contain grooves cut perpendicular to the sliding direction. We now include details of this in the methods section (lines 228-230):

To discourage boundary shear at the edges of the gouge layer, the sliding area (50 × 20 mm) on the forcing blocks contains grooves cut perpendicular to the sliding direction (200 μm deep with 400 μm spacing).

3. Please provide a bit more detail on the jacket. PVC can be used for rigid plumbing pipe. I assume you're using another type and it would be good to provide this information in the supplement.

We use a soft PVC jacket (Nalgene 180 clear tubing) and we have added this detail to the methods section (Line 233):

...placed into a soft, 3 mm thick, PVC jacket (Nalgene 180 clear tubing).

4. Your references are very good. You might also consider commenting on the comparison of your work with that of Niemeijer et al. GRL 2010 (Fabric induced weakness of tectonic faults).

We thank the reviewer for suggesting this reference as it is relevant to our manuscript, particularly regarding the role of weak phase localization on faulting. We have therefore added the reference (ref. number 11) on line 34 and also on line 164.

5. A useful addition to your discussion of Figure 4 would be to connect spatial heterogeneity to the slip patch size for rupture nucleation/propagation. One way to produce slow slip is to limit the size of the velocity weakening region (for example as shown by Liu & Rice 2005/2007) and your data provide some nice connections to these ideas.

Thank you for highlighting that our results relate well to the idea that relative patch size controls the fault slip behaviour in nature. We have added the following text to our discussion of Figure 4 on lines 200-207, although we cited a different example from Liu and Rice (2005/2007) since their studies modified the properties of the velocity-weakening patches more so than the size:

There are similarities between the slip behaviour we observe in our small-scale heterogeneous experiments and how large-scale heterogeneities are thought to control the behaviour of natural faults. For example, decreasing the size of the rate-weakening patch makes the response more stable in both our experiments and numerical modelling⁵³, as can be intuitively expected and consistent with stability studies of rate-and-state faults that slip instability can only result from large enough rate-weakening patches³⁹. At the same time, small-scale fault zone heterogeneity would more readily evolve with shear, and hence may depend on the fault maturity, healing processes, and spatio-temporal history of fault slip.

6. I like the idea of the color bars for strength and seismic potential in Fig. 4 but I'm not sure I see the argument for low strength and high seismic potential. Maybe this is just the same point I raised above, but here you seem to be going beyond the idea of a relative change (heterogeneity promotes unstable slip) to a more definite statement.

We agree with the reviewer and have adjusted the figure so that colour bar now represents “relative seismic potential” between heterogeneous and homogeneous faults. We also now state on the figure that unstable slip requires rate-weakening material:

7. It would be useful to comment on how the kind of heterogeneity you envision would be sustained in a tectonic fault zone with sizable offset. For example, one could imagine that a fault zone with initial heterogeneity of the sort you examine would become homogeneously mixed after a relatively small offset.

Chris Marone

This point was also raised by Reviewer 1 and we have responded in detail there. In part, we have now added additional text on lines 160-173 to discuss the preservation of heterogeneity in natural fault zones:

In our experiments, if the gouge layers could be taken to greater shear displacements, the clay smearing we observe along the edges of the quartz patch (Fig. 2a) would ultimately form a through-going layer of interconnected weak material after a few centimetres of slip. Previous work has shown that such through-going layers can lead to a reduction in the frictional strength at slow slip velocities¹¹ and also increase the efficiency of dynamic weakening at seismic slip velocities (1 m/s)⁴⁴. Although weak phase smearing would, to some extent, homogenize the fault in the overall direction of shear, heterogeneity would likely always be prevalent in natural faults, particularly perpendicular to the slip direction and also at scales larger than investigated in this study, as observed in natural fault zones^{25,45}. Our results show that the average frictional strength of laterally heterogeneous faults is not just an average of the respective friction properties (Fig. 2c), and that competency contrasts can substantially reduce the fault strength, even when structural foliations are in their infancy and unconnected (Fig. 2a). They also highlight the need to investigate further how different types of fault heterogeneity, including fault-parallel and fault-normal heterogeneity, and its evolution, affect the frictional behaviour of faults.

Reviewer 3 comments

Review of 315765-0, "Fault rock heterogeneity produces fault weakness and promotes unstable slip" by Bedford, Faulkner and Lapsuta.

By Terry E. Tullis

General Comments

My comments are both for the editor and the authors. This is a good paper and could be published as is. It is clear and succinctly written and the noteworthy results are made clear in the paper, namely that heterogeneous distribution of rock types along a fault contribute to the strength and stability of the fault in non-intuitive ways and that this is important in modeling earthquakes. It is a significant contribution to the field. The work and methodology is sound and supports the conclusions. Enough detail is presented that the results could be reproduced. However, I have some suggestions for the authors to consider that should not involve much additional work and should improve an already fine paper. I have included these as comments in both the Word file for the article and the Word file for the supplement, using Word's Comments feature under Word's Review tab.

I have copied and pasted below all my marginal comments, but they are better understood in the context of where they appear in the Word doc. Although there are a number of comments scattered among my marginal comments, perhaps the very last one on the Supplement file is the most important, as it affects the reader's understanding of what the authors conclude about the correct explanation for their experimental observations.

Comments in the Article file:

Line 28: mu should be tau

Thank you for finding this mistake, it has now been corrected (Line 29).

Line 129-130: However, if the clay is supporting more of the normal stress and therefore is contributing more than its volumetric proportion to the overall measured properties, why is the velocity dependence seem to be skewed toward the rate weakening quartz? Have you tried to do a quantitative model of a-b as you do for mu? Of course it's not clear how one should to that – I suppose just treat a-b as a parameter with values that go with each phase and that sum up linearly in proportion to their contact area along the two bounding faults as you do for the friction. Not sure if that makes sense or not!

This is an interesting idea and we have now made a quantitative model of (a-b) like we did for mu, by taking the (a-b) values from the endmember gouges (at a displacement of 1.5 mm) and then calculating how these would evolve with displacement as a greater portion of the sliding surface is occupied by clay gouge due to smearing. The (a-b) data are more scattered than the mu data so caution should be taken when interpreting the results, however, the data actually show that clay smearing underpredicts the evolution of (a-b) and that the observed (a-b) values are higher than the predicted values. The data are therefore slightly skewed towards the rate-strengthening clay rather than the rate-weakening quartz. This actually supports our hypothesis that clay is supporting more of the normal stress with increasing displacement, as we hypothesize from the pore volume data. We have now included this quantitative model of (a-b) as a new figure in the supplementary material (Supplementary Fig. 4):

Supplementary Figure 4| Predicted ($a - b$) evolution with displacement from clay smearing in the heterogeneous faults. Also shown are the obtained ($a - b$) values from both heterogeneous and homogeneous fault experiments for clay fractions of **a.** 40%, **b.** 50%, **c.** 60% and **d.** 70%. The predicted ($a - b$) evolution is calculated using the arithmetic mean of the ($a - b$) values for the endmember quartz and clay gouges (from 1.5 mm displacement in Supplementary Table 1) and by assuming the length of the clay patch increases by the amount of displacement on the fault as the clay is smeared along localized Y-shear planes – as was done for coefficient of friction in Fig. 2c of the main manuscript. The ($a - b$) data are more scattered than the friction data in Fig. 2, so caution should be taken when interpreting the results; however, the data show that clay smearing generally underpredicts the ($a - b$) values for the heterogeneous faults. The actual ($a - b$) values from the heterogeneous experiments are higher than predicted, suggesting that the frictional properties of the clay patches contribute more to the average ($a - b$) values, as would be consistent with higher normal stresses in the clay patches than in the quartz patch due to differential compaction which could also potentially explain some of the progressive weakening trends in our experiments (see Supplementary Text below for full discussion of this effect). Note, however, that even though the experimental ($a - b$) values are higher than predicted for the heterogeneous faults, they are still consistently lower than ($a - b$) values from the equivalent homogeneous faults. Also note that only ($a - b$) values from up-steps in the sliding velocity are shown (from 0.3 to $3 \mu\text{m}\cdot\text{s}^{-1}$) as there is asymmetry in the frictional response between velocity up-steps and down-steps (from 3 to $0.3 \mu\text{m}\cdot\text{s}^{-1}$), something that has been reported in previous studies on the rate-dependent frictional behaviour of fault gouges⁵⁵⁻⁵⁷.

Line 135-136: Have you seen any evidence of this in thin sections? Seems as if there might be shear/damage zones in the quartz connecting the tips of the smeared clay on opposite sides of the overall layer. In any case, some comment as to whether this has been observed directly would be helpful as it seems a likely process. Related to this, see my note commenting on Figure 2.

As it is difficult to preserve the gouge layers at the end of the experiment, due to the gouges being largely incohesive, we were unable to collect any detailed microstructural images from the tips of the propagating bands as these were not recovered. We do, however, observe R_1 Riedel shears cross-cutting the quartz patch (an example is visible in Fig. 2a – which we have now highlighted on the figure) which could potentially facilitate weakening by connecting the smeared clay on opposite sides of the layer. We now include additional text to discuss this on lines 151-156:

Due to difficulty keeping the gouge layer intact during recovery at the end of our experiments, we were unable to acquire detailed microstructural images of the tips of the propagating shear bands to look for evidence of shear/damage zones in the quartz patch. We do, however, observe R_1 Riedel shears in the quartz patch (Fig. 2a) which may help facilitate weakening by connecting the smeared clay on opposite sides of the layer.

Line 174-177: This overall problem sort of reminds me of the complexities of deformation of a poly-phase aggregate, where neither the uniform stress or uniform strain models adequately describe the behaviour and some more detailed modelling is needed. Your experimental geometry nominally simpler than that, but even so the simple model is predictions don't seem to adequately explain the results. With patches on a fault plane rather than simple strips perpendicular to the slip direction as in the experiments, the situation will clearly be even more complex.

We agree that our setup is relatively simple and yet it is still difficult to fully explain the behaviour we observe. Our results also highlight the need to perform more experiments to understand better the behaviour and also investigate how other fault geometries (e.g., patches rather than strips) effect the frictional response, which we now mention on lines 171-173.

Line 343-344: I'm thinking that it would be helpful to show calculated curves for 60, 70, and 80 percent clay. Given that the length of the quartz patch along the fault is progressively smaller in those cases, the decline of the calculated curves as a function of slip would occur sooner and sooner as the clay content goes up. This is seen in the data. So looking at this might help to better understand the proportion of the weakening due to clay smearing. This, combined with localized stress concentrations as you discuss on lines 135-136, could help explain the observations.

We agree with the reviewer that Figure 2c would be improved if it showed curves from a greater range of clay fractions. However, we are reluctant to include all the weakening curves in this figure in order to prevent it from becoming cluttered and hard to interpret. We have therefore removed the curves for a clay fraction of 40% and replaced them with weakening curves for the 70% clay sample (i.e., a much smaller quartz patch). The figure now shows weakening curves for samples with 30% and 70% clay fractions, which covers a greater range than our original figure. As can be seen in the figure, the predicted weakening from clay smearing underestimates the overall weakening observed in our experiments over the entire range of clay fractions.

Comments in the Supplement file:

Caption to Supplementary Figure 1: It would help if you give the length of the sample so one is able to think about how the amount of displacement as the experiments progress compares with that length. Without that information one is unable to reproduce your calculations in Figure 2c. Perhaps your figure is to scale and the 20 mm diam. platens is all you need, but the figure could be more schematic. The reviewer is right that the length of the layer helps to interpret this figure; we have now added it into the figure caption (50 mm).

“Similar weakening” in caption for Supplementary Figure 3: I note however that the value of $a-b$ seems to be more positive than in the configuration with the quartz patch in the middle and in fact is similar to the homogeneous gouge (Figure 1d). I presume you have no explanation for this difference, assuming it is significant, which appears to be the case. Either say that you have no explanation for it or maybe you can come up with one. One value of making a comment to this effect is that when some reader notices it you will not seem to be unaware of it! I suggest you include the data for these experiments in Table S1. The reviewer is right that for some of the velocity steps the $(a-b)$ are slightly higher in the reversed symmetry faults, something we hadn’t noticed previously. We have now included those data in Supplementary Table 1 and also added the following text in to the figure caption of Supplementary Figure 3:

Although similar weakening is observed, we note that some of the $(a - b)$ values from the reversed symmetry faults are slightly higher than for the heterogeneous faults comprised of a central quartz patch (Supplementary Table 1). We have no explanation for this difference; although it could be due to $(a - b)$ data from friction experiments generally being quite scattered, when compared to coefficient of friction data, and thus caution should be taken when directly comparing individual velocity steps.

First line of very last paragraph: In spite of these caveats I think it is worth considering an attempt to add another component to Figure 2, i.e. Figure 2d, that includes an attempt to calculate the curves for one or more cases of clay fraction as you do in Figure 2c. I assume you could take the data at each displacement and run through the calculation as a function of displacement. It is always advisable for your figures to tell your story since that is what many readers will focus on. As the paper now stands, Figure 2c leads one to conclude that you don’t really understand the extra observed weakening. Reading the lines near the end of the previous paragraph in the supplement is needed to realize that you actually do have a viable explanation for the weakening. This location hides that understanding too much! The new figure could, as does part c now, only focus on each effect by itself, leaving to words the statement that one looks larger than the other and together they could both clearly do the job.

We thank the reviewer for this constructive comment and we agree that without a demonstration of the differential compaction effect, the figures in our original manuscript do not tell the complete story. We therefore include an extra panel in Figure 2 (Fig. 2d) to show the effects of differential compaction which are significant and can potentially explain the majority of the weakening we observe. We have also added some additional text in the main manuscript to explain this in greater detail (lines 108-118), although we have chosen to keep the majority of the explanation in the supplementary material:

An additional cause of the weakening could be differential compaction between the different gouge materials resulting in a redistribution of normal stress (see Supplementary Information for full discussion of this effect). The volumetric strain data from the endmember quartz and clay gouge experiments show that the quartz gouge experiences a greater layer thickness reduction of about 20 μm than the clay gouge during slip (Supplementary Fig. 2). In the heterogeneous layer experiments this would result in an increase of normal stress on the weaker clay patches leading to a progressive reduction in shear resistance, as observed in our experiments. The magnitude of this effect is dependent on the bulk (K) and shear (G) moduli³⁵, which are poorly constrained for the gouge materials in this study. Using plausible values for the moduli (Supplementary Information) indicates that this differential compaction effect could potentially explain a large component of the weakening we observe in our experiments (Fig. 2d).

Figure 2 | Microstructural evolution and potential causes of weakening in the heterogeneous fault gouge layers. **a**, Backscatter electron image of the interface between a clay-quartz patch recovered at the end of an experiment. The clay phase becomes smeared along a boundary Y-shear plane that propagates into the quartz patch. Since it is difficult to keep the gouge layer intact upon removal from the direct shear assembly at the end of the experiment, the full extent of the localized shear band was not recovered. **b**, Schematic diagram showing the evolution of the fault gouge layers with progressive smearing of the clay phase along localized Y-boundary shears (red box shows the location of the micrograph in **a**). **c**, Observed weakening versus predicted weakening due to clay smearing for heterogeneous layers comprised of 30 and 70% clay fractions. The predicted weakening is calculated using the arithmetic mean of the friction coefficients of the endmember quartz and clay gouges and by assuming that the length of the clay patches increases by the amount of displacement on the fault as clay is smeared along localized Y-shear planes. The observed weakening is considerably greater than the predicted weakening. The labels (I), (II) and (III) correspond to the structural evolution in **b**. **d**, The potential weakening effect from differential compaction between the clay and quartz gouge patches. This effect is dependent on the bulk (K) and shear (G) moduli of the gouge, which are poorly constrained (see Supplementary Information for full discussion). The differential compaction could account for a large component of the weakening in the heterogeneous fault experiments.

REVIEWERS' COMMENTS

Reviewer #1 (Remarks to the Author):

The authors in their rebuttal letter addressed the comments I made and, where they considered to be useful, they discussed them also in the main text.

I find the revised version of the manuscript and of its Suppl. Material ready to be accepted in Nature Communications.

With my best regards,
Giulio Di Toro

Reviewer #2 (Remarks to the Author):

Sorry that it took me a few days to get to this. I've gone through everything and in my opinion the paper can be published as is. The authors did an excellent job making revisions and addressing my concerns/comments and those of the other two reviewers. This is an excellent paper that is appropriate and worthy of being published in Nat. Comm.

Chris Marone

Reviewer #3 (Remarks to the Author):

I am satisfied with the authors response to my original review and have no further recommendations except that the paper be accepted for publication.

Response to reviewers' comments on "Fault rock heterogeneity can produce fault weakness and reduce fault stability"

The reviewer comments on our revised manuscript are shown below in blue. The reviewers were happy with our revisions to the original manuscript and we have therefore made no further changes to the final version. We would like to express our gratitude to all three reviewers for their constructive comments that greatly improved the original version of this manuscript.

Reviewer #1 (Remarks to the Author):

The authors in their rebuttal letter addressed the comments I made and, where they considered to be useful, they discussed them also in the main text.

I find the revised version of the manuscript and of its Suppl. Material ready to be accepted in Nature Communications.

With my best regards,

Giulio Di Toro

Reviewer #2 (Remarks to the Author):

Sorry that it took me a few days to get to this. I've gone through everything and in my opinion the paper can be published as is. The authors did an excellent job making revisions and addressing my concerns/comments and those of the other two reviewers. This is an excellent paper that is appropriate and worthy of being published in Nat. Comm.

Chris Marone

Reviewer #3 (Remarks to the Author):

I am satisfied with the authors response to my original review and have no further recommendations except that the paper be accepted for publication.